# Programming *Escherichia coli* to function as a digital display

Jonghyeon Shin, Shuyi Zhang, Bryan S Der, Alec AK Nielsen & Christopher A Voigt[*] 

## Abstract

Synthetic genetic circuits offer the potential to wield computational control over biology, but their complexity is limited by the accuracy of mathematical models. Here, we present advances that enable the complete encoding of an electronic chip in the DNA carried by *Escherichia coli* (*E. coli*). The chip is a binary-coded digit (BCD) to 7-segment decoder, associated with clocks and calculators, to turn on segments to visualize 0–9. Design automation is used to build seven strains, each of which contains a circuit with up to 12 repressors and two activators (totaling 63 regulators and 76,000 bp DNA). The inputs to each circuit represent the digit to be displayed (encoded in binary by four molecules), and output is the segment state, reported as fluorescence. Implementation requires an advanced gate model that captures dynamics, promoter interference, and a measure of total power usage (RNAP flux). This project is an exemplar of design automation pushing engineering beyond that achievable "by hand", essential for realizing the potential of biology.

**Keywords** design automation; *Escherichia coli*; genetic circuits; logic gates; synthetic biology

**Subject Categories** Computational Biology; Biotechnology & Synthetic Biology

**Mol Syst Biol. (2020) 16: e9401**

## Introduction

Encoding algorithms in DNA would allow cells to be programmed to follow a set of rules or perform problem-solving operations (Khalil & Collins, 2010; Benenson, 2012; Brophy & Voigt, 2014; Purcell & Lu, 2014; Bojar & Fussenegger, 2016). This requires the introduction of synthetic regulatory networks, also known as "genetic circuits", that control when genes are turned on and off. Natural regulatory networks are composed of many interconnected biochemical interactions that have proven difficult to parse and model (Freyre-Gonzalez & Trevino-Quintanilla, 2010; Klipp *et al*, 2016). Genetic circuit design has been aided by the reduction of regulatory functions into gates that perform simple Boolean logic functions (Buchler *et al*, 2003; Kramer *et al*, 2004;

Anderson *et al*, 2007; Cox *et al*, 2007; Rinaudo *et al*, 2007; Tamsir *et al*, 2010; Wang *et al*, 2011; Bonnet *et al*, 2013; Stanton *et al*, 2013; Green Alexander *et al*, 2014; Nielsen & Voigt, 2014; Bradley *et al*, 2016; Nielsen *et al*, 2016; Carr *et al*, 2017; Gander *et al*, 2017; Zong *et al*, 2017; Andrews *et al*, 2018). Genetic design automation software, for example Cello, allows a user to define a desired function, for which algorithms combine gates to build a circuit and encode it in DNA. Central to this software is the quality of the mathematical model describing the gates, which is used to predict how they will perform when connected. This study introduces a gate model that requires little additional parameterization, but captures non-additive effects between input promoters, dynamics, and the utilization of cellular resources that can lead to slow growth and evolutionary breakage. With this model, larger design projects can be undertaken, which we demonstrate by recoding an entire binary-coded digit (BCD) to 7-segment decoder chip.

NOR gates have proven useful in building circuits (Tamsir *et al*, 2010; Gander *et al*, 2017). One design for a transcriptional NOR gate is based on two input promoters in series that drive the expression of a repressor that turns off the output promoter (Fig 1A and B, and Appendix Fig S1). The activities of input and output promoters are reported in relative promoter units (RPUs) as a surrogate for RNA polymerase (RNAP) flux (Materials and Methods; Kelly *et al*, 2009). If the flux is ON from either input promoter, the flux from the output promoter is OFF and vice versa. The NOR gate design simplifies their connection to build a circuit; they can be layered in series by having one's output promoter serves as the input promoter to the next. Predicting how gates can be connected requires knowledge of how the output changes as a function of the inputs(s), captured mathematically as the response function (Yokobayashi *et al*, 2002; Canton *et al*, 2008). The first model for NOR gates simply treated the input as the sum of the RNAP fluxes of the upstream promoters, which then is used as input to a NOT gate response function. Mathematically, the response function for gate $i$ is then written as

$$y_i = \left(y_{\mathrm{max},i} - y_{\mathrm{min},i}\right)\left(\frac{K_i^{n_i}}{K_i^{n_i} + x^{n_i}}\right) + y_{\mathrm{min},i}, \qquad (1)$$

where $x$ and $y$ are the fluxes (RPU) of the input and output promoters, respectively (Kelly *et al*, 2009; Nielsen *et al*, 2016). The

Department of Biological Engineering, Massachusetts Institute of Technology, Cambridge, MA, USA
*Corresponding author. Tel: +1 617 324 4851; E-mail: cavoigt@gmail.com

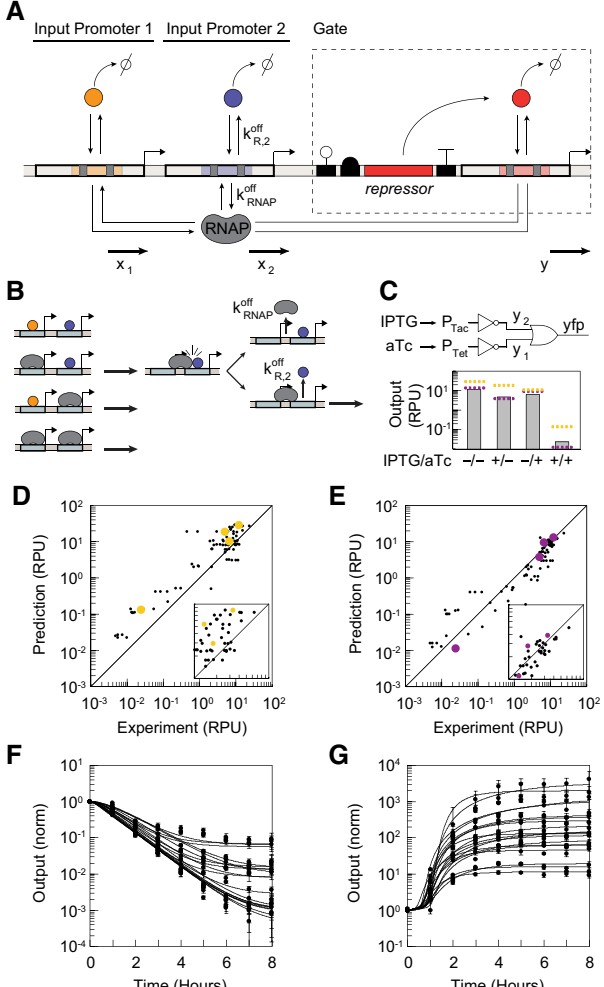

**Figure 1. Dynamic model of NOR gates.**

A A schematic of the NOR gate design is shown. The input promoters are bound by repressors (circles, either from an upstream gate or a sensor). The arrows show the RNAP fluxes entering the gate from the input promoters and flowing out of the gate from the output promoter. The parameters correspond to those described in the text for equations 1–4 and Table 1. The gate symbols correspond to the following (from left to right): ribozyme, ribosome binding site, repressor gene, terminator, and output promoter. Operators in promoters are shown as colors, and shaded regions indicate operator overlap.

B Interference between input promoters. When the repressor is bound to the downstream promoter, this has a probability of blocking RNAPs originating from the upstream promoter. The parameters correspond to those used to calculate β (equation 3). Arrows indicate those promoter states producing RNAP flux.

C An example circuit is shown of the 20 used to evaluate promoters in series. Two NOT gates based on the AmtR (top) and PhlF (bottom) repressors are shown. Their output promoters occupy the upstream (position 1) and downstream (position 2) inputs to the OR gate, respectively. The bar graph shows experimental data for the circuit states (± 1 mM IPTG and 5 ng/ml aTc) compared to predictions based on the additive promoter model (yellow) and model accounting for interference (purple).

D The prediction of the additive model is compared to experiments for the set of 20 circuits constructed (each measured ± 1 mM IPTG and 5 ng/ml aTc). The colored dots correspond to the circuit example in part C. The $x = y$ line is shown ($R^2 = 0.45$).

E As in (D), but data for the model accounting for interference are shown ($R^2 = 0.76$). Note that the experiments in (C–E) were performed using a higher-copy p15a mutant (Materials and Methods). The insets zoom into region corresponding to the H state.

F The dynamic response of each gate for the ON→OFF transition. The two input promoters are turned on by adding 1 mM IPTG and 5 ng/ml aTc (Materials and Methods). Equation 4 was fit to the data for each gate (Table 1). Eighteen gates are shown; individual responses are provided in Appendix Fig S7. The responses are normalized by the value at $t = 0$.

G The same as in part F for the OFF→ON transition.

Data information: The data show the average of three experiments performed on different days, and the error bars represent the standard deviation.

parameters $y_{max}$, $y_{min}$, $K$, and $n$ are all determined empirically. The additive approximation assumes that the RNAP fluxes from the two upstream promoters can be summed, $x = x_1 + x_2$.

## Results and Discussion

A set of simple circuits were constructed to test the additive approximation. Building on previous studies, a set of six NOT gates and a 3-oxo-N-[(3S)-tetrahydro-2-oxo-3-furanyl]-hexanimide (OC6) sensor were genetically optimized to increase their dynamic range and lower their OFF state, thus making them easier to connect (Materials and Methods, Appendix Figs S2 and S3). These were used together with IPTG and aTc sensors to build 20 circuits that function as NAND gates, comprised of two NOT gates that connect to an OR gate (Fig 1C). The circuits were then induced with different combinations of inputs and compared with that predicted using the response functions of the two NOT gates ($y_1$ and $y_2$) (equation 1) and the additive approximation for the promoters in series serving as the OR gate. An example of such a prediction is shown in Fig 1C (yellow lines) and compared to experimental data obtained when the circuit is constructed. This was repeated for the complete set of 20 circuits

(20 × 4 = 80 predictions), and the data are plotted in Fig 1D and Appendix Fig S4.

Promoters in series, even when spatially separated, have the potential for transcriptional interference, where one has a suppressive effect on the other (Sneppen et al, 2005; Hao et al, 2017; Zong et al, 2018). Notably, when a repressor is bound tightly to its operator and has a slow off-rate relative to the RNAP elongation rate, RNAPs originating upstream may not be able to dislodge it, referred to as "roadblocking" (Deuschle et al, 1986; Sellitti et al, 1987; Ahlgren-Berg et al, 2016). The challenge with gate design is that those features that make a good gate (e.g., a low OFF state and cooperativity) also make its output promoter more prone to roadblocking (Fig 1A and B, and Appendix Fig S1). Previously, we addressed this in a draconian way, where a promoter's proclivity to roadblock was measured empirically and those that passed a threshold were excluded from occupying the downstream position (Position 2 in Fig 1A; Nielsen et al, 2016). This greatly reduces the potential solutions in assigning gates to a circuit and, because all repressor-bound promoters roadblock to some degree, leads to inaccurate predictions. In practice, this restriction leads to Cello being unable to find solutions for some circuit designs (Appendix Fig S5).

We hypothesized that interference between the input promoters is a contributor to the inaccurate predictions in Fig 1D. To address this, a new gate model was developed that incorporates roadblocking interactions (Fig 1B). Roadblocking occurs when RNAP is unable to dislodge the repressor, thereby reducing transcription from the upstream promoter. The model also allows for interference to occur via repressor-independent mechanisms (pause sites, antisense transcription, binding of other proteins, etc.) (Dahirel *et al*, 2009; Roberts, 2014; Brophy & Voigt, 2016). From a detailed model that captures these effects, a correction to the response function of the upstream promoter can be derived (Appendix Supplementary Text),

$$y_1 = \alpha \left( \frac{K_2^{n_2} + \beta x_2^{n_2}}{K_2^{n_2} + x_2^{n_2}} \right) \left[ (y_{\max,1} - y_{\min,1}) \left( \frac{K_1^{n_1}}{K_1^{n_1} + x_1^{n_1}} \right) + y_{\min,1} \right], \tag{2}$$

where the binding of the repressor to input promoter 2 impacts the response function of input promoter 1 (square brackets). The non-specific suppression of the upstream promoter is captured by the parameter $\alpha$. The degree of roadblocking is represented by the first term in parenthesis, which is dependent on the repressor binding to the downstream promoter and includes the parameter

$$\beta \equiv \frac{k_{R,2}^{\text{off}}}{\frac{x_1}{[\text{DNA}]} + k_{\text{RNAP}}^{\text{off}} + k_{R,2}^{\text{off}}}, \tag{3}$$

where $x_1/[\text{DNA}]$ is the RNA flux from a single upstream promoter. Equation 3 captures competing effects between the kinetics of the repressor off-rate $k_{R,2}^{\text{off}}$ and dissociation rate $k_{\text{RNAP}}^{\text{off}}$ (Hao *et al*, 2014). The parameters $\alpha$ and $\beta$ are both dimensionless and are treated as fit parameters. Smaller values of both indicate more interference, and as $\alpha$ and $\beta$ approach unity, the interference effects go to zero. The output promoter for sensors can also roadblock, and we derive similar equations to derive these parameters (Appendix Equations S10 and S13).

An experimental approach was developed to determine the parameters $\alpha$ and $\beta$ of the output promoter associated with each gate. The challenge is that we did not want to test a promoter with all possible upstream promoters for a set of gates, as this does not scale well for larger gate libraries. Instead, we assume that the parameters are independent of the upstream promoter identity. This assumption is reasonable because all of the gate's output promoters are based on a shared simplified promoter architecture and exhibit similar dynamic ranges (Epshtein *et al*, 2003; Hao *et al*, 2014; Nielsen *et al*, 2016). For the evaluation, the aTc-inducible $P_{\text{Tet}}$ promoter is placed in the upstream position. The IPTG-inducible $P_{\text{Tac}}$ promoter is used as the input to the NOT gate being evaluated. Four data points are taken for the combination of the presence and absence of inducers. These data are fit to equation 2 from which the $\alpha$ and $\beta$ parameters are extracted (Table 1, Appendix Fig S6). Note that the degree of roadblocking is promoter-dependent and differs significantly across promoters, thus making it impossible to assign a single "correction factor" equally to all promoters. The most roadblocking gates (PhlF, SrpR, BM3RI, CymR) are also the most cooperative ($n = 3.9$, 2.9, 3.3, and 3.7), whereas other measures of repressor activity ($K$, $y_{\min}$) are not predictive. The model

incorporating interference was then used to predict the states associated with the test set of 20 circuits (Fig 1E and Appendix Fig S4). A similar approach is taken to parameterize the sensor output promoters (Appendix Fig S7).

The response functions only describe a gate's behavior at steady state. Kinetic models of gene regulation often require many parameters that are difficult to empirically measure. Instead, we took the approach of using a simplified model that only has two parameters ($\tau_y^{\text{ON}}$ and $\tau_y^{\text{OFF}}$) that describe the timescale by which a gate turns on or off. This approach models gate dynamics using an ordinary differential equation (ODE)

$$\frac{dy}{dt} = \begin{cases} \tau_y^{\text{ON}}(y_{\text{ss}} - y) & \text{if } y < y_{\text{ss}} \\ \tau_y^{\text{OFF}}(y_{\text{ss}} - y) & \text{otherwise} \end{cases}, \tag{4}$$

where $y$ is the output and $y_{\text{ss}}$ is the steady-state response, defined by the states of the inputs. This equation captures the different mechanisms underlying the return to steady state when the gate output is higher or lower than the steady-state value. If it is lower, then the response is dominated by repressor degradation, whereas if it is higher, it is dominated by transcription and translation.

Two sets of experiments were performed to obtain $\tau_y^{\text{ON}}$ and $\tau_y^{\text{OFF}}$ for each gate. First, ON→OFF experiments are performed where cells containing the gate are grown in the absence of inducers until reaching steady state (Materials and Methods). Then, the cells are transferred to fresh media with inducers and time points are taken as the gates relax to the OFF state (Fig 1F and Appendix Fig S8). All of the gates switch off following a similar exponential decay. From these data, $\tau_y^{\text{OFF}}$ can be determined, noting that the response of the sensors and the production of YFP have to be incorporated into the model (Table 1, Appendix Fig S9, Appendix Information II). Next, to obtain OFF→ON switching dynamics, cells are grown in the presence of inducers until the output promoters of the gates are fully repressed. The cells are then transferred to fresh media lacking inducers, and time points are taken as the gate turns on (Fig 1G and Appendix Fig S8).

In Cello, the "user constraint file" (UCF) defines the species, genetic location, and gate technology that is used by the software to design a circuit. The first UCF was for *Escherichia coli* for the design of circuits in the p15a plasmid backbone (Eco1C1G1T1). In this work, a new UCF was created based on the optimized gates and parameters, including the non-additive promoter inputs and dynamics (Eco1C2G2T2, Dataset EV1). The restrictions on assigning an input promoter to Position 2 because of roadblocking were removed. A user specifies a desired circuit function using the Verilog language and specifies the sensors (including their ON/OFF response in RPU) and the UCF. Cello then loads the UCF, creates a wiring diagram for the specified circuit function, assigns repressors to each gate according to their response functions, and then maps the circuit to a linear DNA sequence (Materials and Methods). The software uses the response functions of the gates to predict the circuit response. To predict cell-to-cell variation due to intrinsic and extrinsic noise, the software uses cytometry data from the UCF describing the responses of each gate to predict the overall population response of the circuit. Finally, the growth impact of each gate (measured as $OD_{600}$) as a function of the flux from the input promoter is used to predict the overall growth impact of carrying the circuit in different states.

**Table 1.  Parameterization of gates.**

| | Response function[a] | | | | Promoter interference[a] | | Kinetics[a] | |
|---|---|---|---|---|---|---|---|---|
| | $y_{min}$ | $y_{max}$ | $K$ | $n$ | Non-specific ($\alpha$) | Roadblocking ($\beta$)[b] | Induction ($\tau_y^{ON}$) | Relaxation ($\tau_y^{OFF}$) |
| F1-AmeR_2 | 0.29 | 4.6 | 0.12 | 1.5 | 0.09 | 1.00 | 8.00 | 2.30 |
| F2-AmeR_2 | 0.21 | 3.7 | 0.04 | 1.3 | 0.10 | 1.00 | 8.00 | 2.50 |
| N1-LmrA_2 | 0.077 | 1.3 | 0.09 | 1.8 | 0.28 | 1.00 | 1.00 | 1.30 |
| A1-AmtR_2 | 0.035 | 3.1 | 0.05 | 1.7 | 0.27 | 1.00 | 0.90 | 2.50 |
| H1-HlyIIR_2 | 0.004 | 2.1 | 0.13 | 2.6 | 0.15 | 1.00 | 0.45 | 4.00 |
| P1-PhlF | 0.004 | 6.9 | 0.04 | 3.8 | 0.22 | 0.06 | 0.30 | 4.00 |
| P2-PhlF | 0.007 | 7.5 | 0.21 | 4.5 | 0.15 | 0.12 | 0.20 | 4.00 |
| P3-PhlF | 0.004 | 7.1 | 0.12 | 3.3 | 0.24 | 0.06 | 0.15 | 5.00 |
| S1-SrpR | 0.001 | 1.4 | 0.02 | 3.1 | 0.50 | 0.05 | 5.00 | 7.00 |
| S2-SrpR | 0.003 | 3.2 | 0.06 | 2.7 | 0.35 | 0.09 | 0.50 | 7.00 |
| S3-SrpR | 0.004 | 3.1 | 0.09 | 2.7 | 0.33 | 0.10 | 0.70 | 6.00 |
| S4-SrpR | 0.004 | 3.2 | 0.11 | 2.7 | 0.38 | 0.09 | 1.80 | 8.00 |
| E1-BetI_2 | 0.041 | 2.8 | 0.28 | 2.9 | 0.64 | 0.46 | 0.40 | 1.50 |
| B1-BM3R1 | 0.004 | 0.6 | 0.06 | 2.9 | 0.64 | 0.05 | 0.90 | 1.10 |
| B2-BM3R1 | 0.006 | 0.5 | 0.57 | 3.8 | 0.71 | 0.05 | 0.90 | 1.10 |
| B3-BM3R1 | 0.005 | 0.6 | 0.21 | 3.1 | 0.74 | 0.05 | 0.50 | 2.00 |
| C1-CymR | 0.010 | 3.0 | 0.10 | 3.7 | 0.10 | 0.07 | 0.50 | 2.50 |
| V1-VanR | 0.043 | 6.2 | 0.05 | 2.9 | 0.12 | 0.32 | 0.40 | 11.0 |

[a]The maximum allowed value is one during fitting.
[b]The units of $y_{min}$ and $y_{max}$ are RPU. The units of $K$ are RPU. The timescales are in units of 1/h. $n$, $\alpha$, and $\beta$ are unitless.

A textbook example of logic design was selected to demonstrate the advanced gate model (Marcovitz, 2004; Floyd, 2011). Electronics, especially old calculators, used a liquid crystal display (LCD) where the digits 0–9 could be displayed by turning on different patterns of a set of seven fixed segments (A–G, Fig 2A). A chip (e.g., Texas Instruments SN74LS49, first introduced in 1974) served the role of receiving the number to be displayed as a binary-coded decimal (four inputs to represent 0–9) and calculating which of the seven segments to display as outputs. Layered logic gates in the chip perform the calculation (Fig 2B).

We envisioned that each segment could be formed by a strain of *E. coli* carrying a 4-input logic circuit to determine when to turn on a reporter to form a digit (Fig 2C). The desired digit is communicated to the cells using small molecules associated with four inducible systems (OC6, aTc, OHC14, and IPTG), and the presence/absence of the molecule represents the 1/0 of a binary code (e.g., 3 = 0011 = −IPTG/−OHC14/+aTc/+OC6). The complete design requires seven large circuits containing a total of 63 regulatory proteins, encompassing seven gates and four sensors, that need to interact precisely so that the cells respond properly across $7 \times 2^4 = 112$ states. The scale of this project was outside of what is possible with the previous gate model, which was unable to compute solutions for some circuits, and others were predicted to function inadequately. The results of running Cello with the previous UCF (Eco1C1G1T1), including where a design solution could not be found, are shown in Appendix Fig S5.

The seven circuits corresponding to each segment were designed using Cello and the new UCF (Eco1C2G2T2) with no additional optimization of the DNA sequence (Fig 2C and D, and Appendix Fig S10). The DNA for each circuit was constructed, transformed into *E. coli*, and evaluated for the correct logic function (Materials and Methods). All circuits functioned properly across the full set of 112 states with an average of 260-fold difference in output between the OFF and ON states (Fig 2E and Appendix Fig S11). There is a good separation of OFF and ON states, with the worst being a 7-fold difference between the highest OFF and lowest ON states exhibited by segment B. The population-level predictions matched the cytometry data closely (Appendix Figs S10 and S12).

The strains were then tested in the context of acting as segments in a "display" (Fig 3A). To visualize the circuit performance, we made a 3D-printed scaffold containing chambers that mimic the pixels of a LCD display (Fig 3A). Each chamber contains a 20 μl aliquot of the cells containing the circuits. The cells are grown separately and transferred into the segments to a final concentration calculated so that each circuit's fluorescence in the maximum ON state is equal (see caption of Fig 3A and Materials and Methods). The strains are instructed to produce a digit by adding the corresponding combination of the four inducers in culture. Under white light, the bacteria arranged in seven segment patterns look identical and are reminiscent of the digital display of a calculator that is off. The fluorescence image clearly shows all ten digits (Fig 3A). Differences in the signal intensity occur because of different levels of absolute fluorescence corresponding to ON states (Appendix Fig S10).

Long time courses were performed where cells were continuously propagated and switched through all 10 digits in order. Each time that the cells were switched, they were washed and

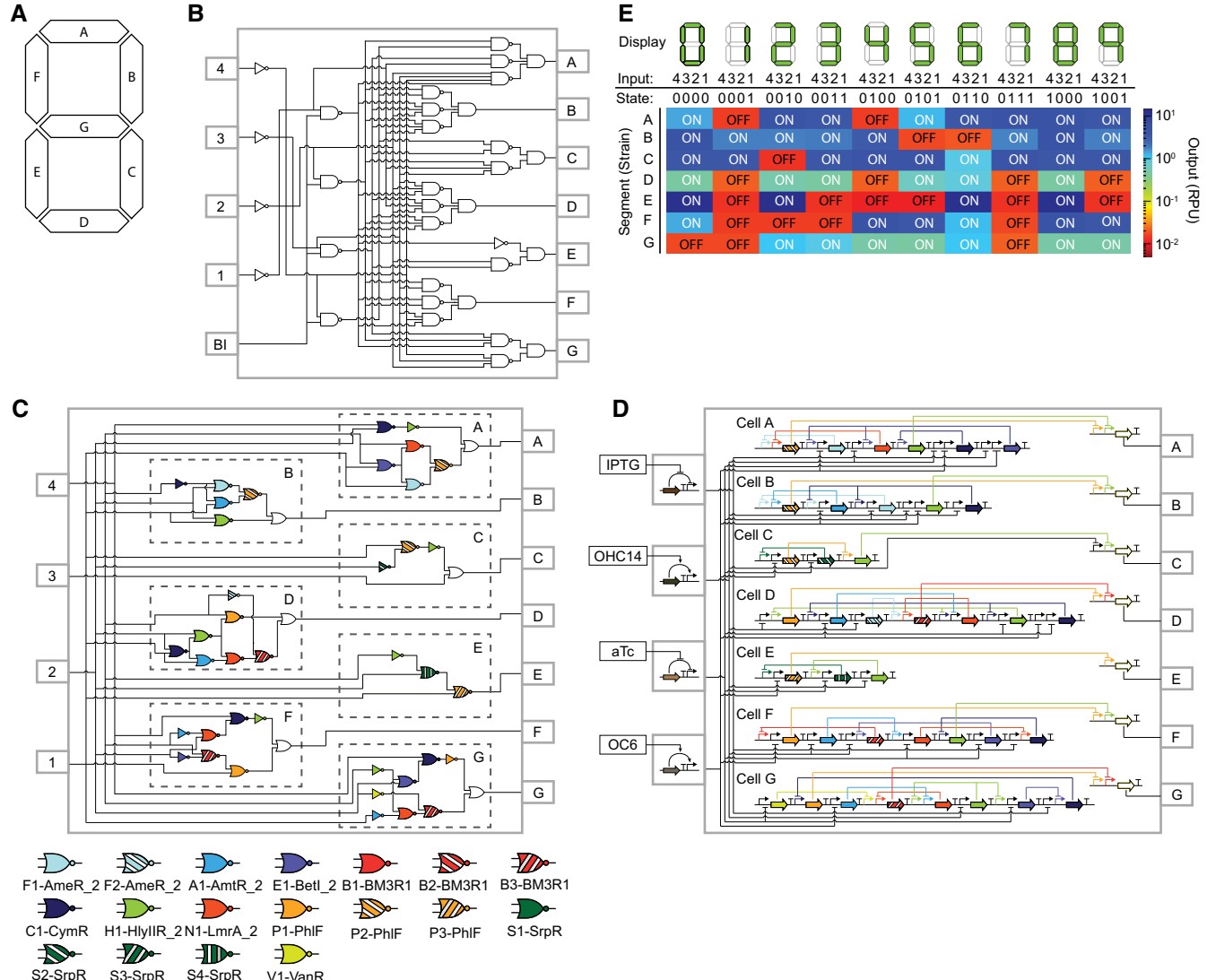

**Figure 2. Genetic circuits that encode the BCD-to-7-segment decoder.**

A   Layout of 7-segment display.

B   BCD-to-7-segment decoder and logic diagram for an electronic circuit. The digit is read as a binary (inputs 1–4), and the outputs corresponding to the segments are shown (A–G). The circuit is based on NOT/AND/NAND gates, and BI is the blanking input (if it is 1, then the segment is displayed; this is not relevant for the design of genetic circuits).

C   The logic circuits associated with each segment are shown. The assignment of gates to each circuit identified by Cello is shown.

D   The mapping of circuits to DNA sequences is shown. The gene colors for the repressors correspond to the gate colors. Each circuit is carried in a different strain (cells A–G).

E   The performances of the segment circuits. Each digit requires different segments to be on (green in top row). The digit encoded as a binary is shown (e.g., 4 in binary is 0100 meaning that only input 3 is on and the other inputs are off). The empirical performance of each circuit is shown (the mean of three cytometry experiments, performed on different days). An example of cytometry plots and the fits to the response predicted by Cello are shown in Appendix Fig S10. Inputs 4/3/2/1 are $P_{Tac}$ (0.2 mM IPTG)/$P_{Cin}$ (1 μM OHC14)/$P_{Tet}$ (2 ng/ml aTc)/$P_{Lux2}$ (0.1 μM OC6). ON and OFF reflect those states that should be on (blue) or off (red) for that combination of inputs.

resuspended in fresh media with the new combination of inducers (Materials and Methods). Including the initial cell growth in the 0000 state for 8 h, this is an 88-h experiment, where cells undergo approximately 130 cell divisions. Each time course was performed in triplicate starting on different days (Fig 3B). The switching dynamics were simulated using the kinetic parameters and found to match the experiments closely (Fig 3B, Appendix Fig S12, and

Appendix Information II). The circuit designs shown in Figs 2 and 3 are stably maintained in the bacteria over multiple days and growth phases while being transitioned between states. Indeed, the only genetic mutation we observed was the insertion of a transposable element into the segment F circuit, discovered at the end of the time course (it did not impact the output during the long experiment) (Appendix Fig S13).

However, *en route* to these designs, we built an earlier generation of segmentation circuits that showed significant problems with genetic stability (Appendix Figs S14 and S15). While optimizing the gate responses, the plasmid backbone picked up a spurious mutation that increased its copy number by 5-fold. This was not corrected because the resulting gates showed superior performance, likely because of higher repressor expression that leads to larger dynamic ranges. Based on these data, a different UCF was created for the higher-copy backbone, and this was used to design circuits using Cello. The resulting circuits functioned properly in culture, but showed state-dependent growth defects and sensitivity to environmental conditions. Troublingly, during longer experiments they would pick up mutations to the plasmid causing them to fail (Appendix Figs S14 and S15). Ultimately, we decided to restart the project by redesigning the underlying gates and using them to rebuild the circuits from scratch.

We hypothesized that the higher propensity for failure was caused by an increased drain on cellular resources, including RNAP and ribosomes. The model can be used to predict the amount of RNAP required by the circuit at a particular time by adding up the RNAP fluxes from all the promoters ($J_{RNAP}$) (Fig 3C). This total RNAP flux is analogous to the power requirements of an electronic circuit. When a genetic circuit requires more power, this is drawn from the limited number of RNAPs of the host, thus reducing the power available to drive host processes (Del Vecchio *et al*, 2008, 2018; Gyorgy & Vecchio, 2014; Qian *et al*, 2017). This also corresponds to higher usage of other resources, including ribosomes, ATP, and amino acids (Scott *et al*, 2010). Collectively, this leads to evolutionary pressure to genetically break the circuit (Canton *et al*, 2008; Tan *et al*, 2009; Sleight *et al*, 2010; Bonnet *et al*, 2013; Chen *et al*, 2013; Jayanthi *et al*, 2013; Sleight & Sauro, 2013; Klumpp & Hwa, 2014; Mishra *et al*, 2014; Ceroni *et al*, 2015; Borkowski *et al*, 2016; Ellis, 2019). The power requirements of a circuit change in each state because different patterns of promoters are active. It has been previously observed that the genetic stability of a sensor is related to whether it is on or off (Canton *et al*, 2008) and the growth impact of a circuit is state-dependent (Fernandez-Rodriguez *et al*, 2015; Lee *et al*, 2016; Nielsen *et al*, 2016; Moser *et al*, 2018). Using this approach, $J_{RNAP}$ can be quantified for the first (genetically unstable) and second (genetically stable) designs of the circuits. Breakages most often occurred when higher $J_{RNAP}$ is required (diamonds, Fig 3C). The second designs of the circuits consistently require less power at all times.

There are several ways that resource usage could be incorporated into circuit design to avoid slower growth and evolutionary breakage. First, gates could be designed to minimize resource requirements. This could be accomplished by encoding them at lower copy in the genome, using high-affinity repressors to avoid overexpression, and using smaller repressors. Second, circuits could be designed to reduce the worst impact of a state or to be carried the longest in a low-impact state. There are many ways that a desired function can be converted to a wiring diagram and one could be selected for reduced impact, rather than minimal size. Third, the algorithms could incorporate the total RNAP flux as part of the objective function and provide a warning if the circuit has states that are above this threshold. While we created a set of circuits in this manuscript that appear to be on either side of this threshold, we do not have enough data to define it definitively. Beyond circuit

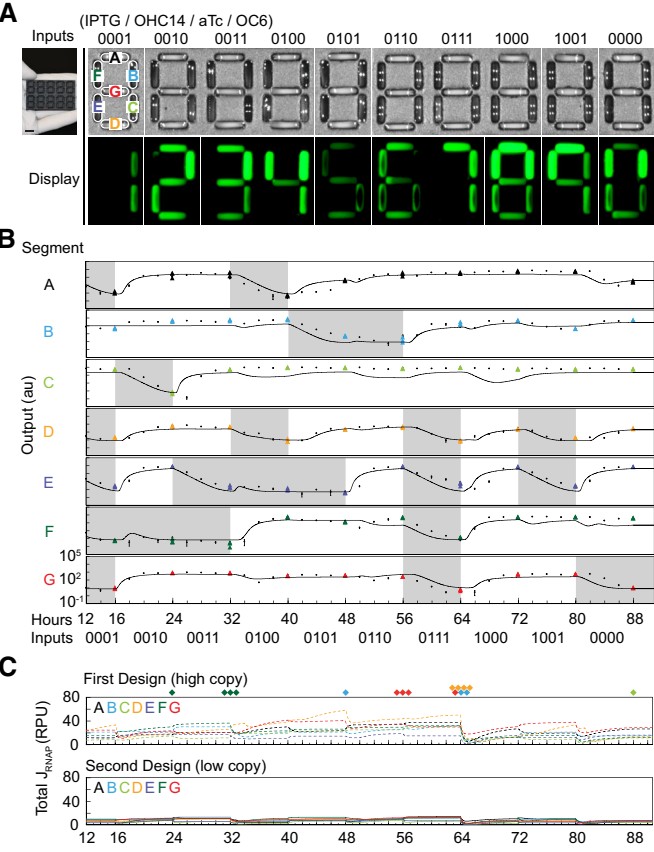

**Figure 3. Performance of the 7-segment circuit.**

A  *Escherichia coli* displaying the digits 0–9. Cells containing each circuit (A–G) are induced and loaded into the well marked with that letter (Materials and Methods). The 3D-printed display is shown in the inset, and a high-resolution image is provided in Appendix Fig S16. The top row shows the sensors that are on for each of the digits (inputs are in order 4/3/2/1 and correspond to $P_{Tac}$ (0.2 mM IPTG)/$P_{Cin}$ (1 μM OHC14)/$P_{Tet}$ (2 ng/ml aTc)/$P_{Lux2}$ (0.1 μM OC6). The quantity of bacteria added to each well is as follows: 0.77, 1.22, 0.13, 1.16, 0.18, 0.88, and 0.80 (×10⁹ CFU/20 μl) (cells A–G). This is to account for variation in the magnitude of the OFF/ON states of the different circuits (see Fig 2F).

B  Dynamics of circuit performance as they are transitioned from 0 → 1 → 2 → 3 ... 8 → 9 → 0. Colored triangles correspond to time points taken every 8 h over a continuous time course (Materials and Methods). The black points show individual experiments that start at one digit and stop at the next to fill in points between transitions at 2-h resolution. Gray regions in the trajectory show those portions of the time course where the output of that segment should be in the OFF state to make the correct digit. The line shows the predicted response based on a set of ODEs and the kinetic parameters for the sensors, gates, and YFP production (Appendix Information). The data points are the average of three experiments performed on different days, and the error bars are the standard deviation of these measurements.

C  The total flux of RNAP is calculated for all of the promoters in the genetic circuit using the dynamic model. The calculation is performed over the 88-h time course, switching between digits as above. "High Copy" refers to an earlier design for the 7-segment circuits based on a mutant backbone that increased copy number (see text and Appendix Fig S14 for details). "Low Copy" refers to the circuits shown in Fig 2. The diamonds at the top of the graphs show when the circuit failed to produce the correct output to make the desired digit (Appendix Fig S14).

design, it would be impactful to genetic engineering projects if there were simple metrics of resource utilization assigned to parts and devices that could be used to predict the stability of a synthetic genetic system prior to it being constructed (Jack *et al*, 2015).

Implementing the BCD converter required balancing many regulatory proteins such they dynamically work together as a network to perform the desired function. Core to our approach is the characterization of individual sensors and gates in isolation such that the information can be used to inform how they can be combined in new ways. Earlier work to insulate gates and simplify their connection enabled circuits to be designed with software (Nielsen *et al*, 2016). Scaling to larger designs required more precise models and a means to predict the impact on the host. Currently, more accurate models are limited by the use of plasmids that result in poorly characterized copy number distributions, retroactivity due to gates sharing resources, non-additive effects due to growth inhibition, and the use of fluorescent proteins as diagnostics. These problems could be addressed by moving circuits to the genome and characterizing circuits using deep sequencing methods to parameterize detailed transcription and translation. Taking on circuit construction, design challenges of increasing size will drive the development of next generation of gates with improved response times, lower power requirements, and higher reliability. Ultimately, this cycle of pushing the design limit, identifying failure modes, and then creating improved gates and models will lead to large-scale integrated circuits in DNA that control cells as autonomous therapeutic agents, architects of functional nanomaterials, and environmental sentinels in agriculture.

# Materials and Methods

### Strain, media, and inducers

*Escherichia coli* strain NEB 10-beta [Δ*(ara-leu) 7697 araD139 fhuA ΔlacX74 galK16 galE15 e14- φ80dlacZΔM15 recA1 relA1 endA1 nupG rpsL (Str^R) rph spoT1 Δ(mrr-hsdRMS-mcrBC)*] was used for all cloning and experiments (New England Biolabs, C3019). Cells were grown in either LB (Miller, BD Difco, 244620) or M9 minimal media containing M9 minimal salts (6.78 g/l Na$_2$HPO$_4$, 3 g/l KH$_2$PO$_4$, 1 g/l NH$_4$Cl, 0.5 g/l NaCl; Sigma-Aldrich, M6030), 0.4% D-glucose (Fisher Chemical, D16-1), 0.2% casamino acids (BD Bacto, 223050), 0.34 g/l thiamine hydrochloride (Sigma-Aldrich, T4625), 2 mM MgSO$_4$ (Affymetrix, 18651), and 0.1 mM CaCl$_2$ (Sigma-Aldrich, C1016). Antibiotics used to select and maintain plasmids were 100 μg/ml carbenicillin (Gold Biotechnology, C-301) and 50 μg/ml kanamycin (Gold Biotechnology, K-120). Chemical inducers used as inputs were isopropyl β-D-1-thiogalactopyranoside (IPTG; Sigma-Aldrich, I6758), anhydrotetracycline hydrochloride (aTc; Sigma-Aldrich, 37919), L-arabinose (Ara; Sigma-Aldrich, A3256), N-(3-Hydroxytetradecanoyl)-DL-homoserine lactone (OHC14; Sigma-Aldrich, 51481), and N-(β-Ketocaproyl)-L-homoserine lactone (OC6; Sigma-Aldrich, K3007). The reporter gene used for these experiments is yellow fluorescent protein (Appendix Table S3; Cormack *et al*, 1996).

### Genetic changes to optimize gate function

Six of the gates from the first library (Nielsen *et al*, 2016) were modified to improve performance (increase dynamic range and

lower background). The gates published previously and unmodified in this work were re-characterized and parameterized together with the modified gates in order to create a dataset under uniform conditions. The gate modifications were made using the JS_BB_1 plasmid backbone (Appendix Fig S17) (Nielsen *et al*, 2016). For the AmeR gate, we modified the RBS (F1) controlling repressor gene expression. To do this, the F1-AmeR gate plasmid was amplified by PCR with divergent abutting primers containing degenerate nucleotides in and around the gate's RBS (5′-AAACATAGTCCATAGCGTATT AAACAAAATTATTTGTAGAGGG-3′ and 5′-CAAACASGMGCTAAT AGATGAACAAAACCATTGATCAGGTGCGTAAAG-3′), and the PCR products were ligated by T4 DNA ligase (NEB, M0202S) and T4 polynucleotide kinase (NEB, M0201L). The mutants were transformed into *E. coli* NEB 10-beta, and several hundred colonies were picked and grown in M9 media in the absence (ON) and presence (OFF) of 1 mM IPTG. After 24 h of incubation on a LB-agar plate, 96 colonies were picked and grown in 150 μl LB media in a V-bottom 96-well plate (Nunc, 249952) at 37°C at 1,000 rpm in an ELMI shaker (ELMI, DTS-4) overnight (15–16 h). The overnight culture was diluted 200-fold into 200 μl M9 media in a V-bottom 96-well plate and grown at 37°C at 1,000 rpm in an ELMI shaker for 3 h. The 3-h culture was diluted twice: 7-fold dilution by adding 10 μl into 60 μl fresh M9 media, and then 100-fold dilution by distributing 2 μl of 7-fold diluted culture into 198 μl M9 media in the absence (ON) and presence (OFF) of 1 mM IPTG. After 7 h of incubation, cells were prepared for cytometry measurement. The top five mutants showing the highest ON to OFF ratios of fluorescence by cytometry were selected for a more exhaustive screening of induction (0, 0.005, 0.01, 0.02, 0.03, 0.04, 0.05, 0.07, 0.1, 0.2, 0.5, and 1 mM IPTG) to identify those with: (i) a desirable ON to OFF threshold consistent with the other gates in the library, (ii) a high dynamic range, and (iii) a low OFF state (Appendix Fig S2). For the F2-AmeR, A1-AmtR, E1-BetI, H1-HlyIIR, and N1-LmrA gates, a promoter library was screened to reduce the OFF state (leakage) when the gate is fully induced. Gate plasmids were entirely amplified by PCR with divergent abutting primers containing degenerate nucleotides in and round a promoter region (F2-AmeR: 5′-NNTGCTAGCAGCTGTCA CCG-3′ and 5′-NNACCTAGGAAGTGATGAGTTGTCA-3′, A1-AmtR: 5′-NNAGTTTCTATCGATCTATAGATAATGCTAGC-3′ and 5′-NNA TTGGTAACGAATCATTTGGTT-3′, E1-BetI: 5′-NNAATTGATTGGAC GTTCAATATAATG-3′ and 5′-NNATTGGTAACGAATCCCTCTCA-3′, H1-HlyIIR: 5′-NNATATTTAAAATTCTTGTTTAAAATGCTAGC- 3′ and 5′-NNATTGGTAACGAATCGTTCAGATT-3′, N1-LmrA: 5′-NNAACT GGTGGTCGAATCAAGA-3′ and 5′-NNATTGGTAACGAATCAGACC TAGTG-3′) and the PCR products were ligated by T4 DNA ligase and T4 polynucleotide kinase. The mutants were screened in two steps, as described above for the RBS library. Gates with a modified output promoter relative to the first library are designated with a "_2" (Appendix Fig S2). All genetic part sequences are provided in Appendix Table S4. Note that some mutations in the top clones occurred inside the targeted part sequence but outside of the degeneracy regions of the primers used for plasmid amplification.

### Genetic change to the OC6 sensor

The $P_{Lux}^*$ output promoter was modified to improve the OC6 sensor. This modification was made using the JS_BB_2 backbone (Appendix Fig S17). This promoter is a mutant of the original $P_{Lux}$

promoter where the −10 box was mutated to reduce the OFF state (leakiness) (Moon *et al*, 2012). However, this mutation also decreased the ON state. To correct this, several primers were designed that made mutations to the −10 box and the entire plasmid was amplified by PCR. The mutants were screened in the presence (ON) and absence (OFF) of 1 μM OC6 under the conditions described in the previous section. The mutant with the highest dynamic range corresponded to the primer pair: 5′-TTTTCGAA TAAAAGCTGTCACCGGATG-3′ and 5′-TAACAAACCATTTTCTTGCG TAAACCTG-3′. This promoter is designated $P_{Lux2}$ (Appendix Fig S3 and Appendix Table S4).

### Sensor characterization

Plasmid maps are provided in Appendix Fig S19 (JS_in_1, JS_in_2, JS_in_3, and JS_in_4). A single colony containing a sensor plasmid was inoculated into 150 μl LB media in a V-bottom 96-well plate (Nunc, 249952) and grown at 37°C at 1,000 rpm in an ELMI shaker (ELMI, DTS-4) overnight (15–16 h). The overnight culture was diluted 200-fold into 200 μl M9 media in a V-bottom 96-well plate and grown at 37°C at 1,000 rpm in an ELMI shaker for 3 h. Then, the $OD_{600}$ of each sample was measured in a plate reader (Synergy H1 microplate reader, Biotek). The samples were diluted to $OD_{600} = 0.036$ into fresh M9 media and sequentially diluted to $OD_{600} = 0.00036$ by distributing 2 μl of the first diluted sample into 198 μl M9 media supplemented with inducers. Cells were grown at 37°C at 1,000 rpm in an ELMI shaker for 8 h before samples were prepared for cytometry.

### Flow cytometry measurement and analysis

Cell aliquots were diluted in phosphate-buffered saline (PBS; OmniPur, 6505-OP) containing 2 mg/ml kanamycin. Fluorescence was measured using the LSRII Fortessa flow cytometer (BD Biosciences). At least 30,000 events were collected and analyzed using FlowJo (TreeStar). The median value of the fluorescence distribution was converted to relative promoter units (RPUs) using the protocol and pJSBS_RPU_standard plasmid (BBa_J23101 promoter and p15A origin) described previously (Andrews *et al*, 2018).

### Tandem promoter experiments

The following describes the experiments corresponding to the results shown in Fig 1C–E (plasmids JS_NOR_101—120, Appendix Fig S23). A single colony was inoculated into 150 μl of LB media in a V-bottom 96-well plate (Nunc, 249952) and grown at 37°C at 1,000 rpm in an ELMI shaker (ELMI, DTS-4) overnight (15–16 h). The overnight culture was diluted 200-fold into 200 μl M9 media in a V-bottom 96-well plate and grown at 37°C at 1,000 rpm in an ELMI shaker for 3 h. Then, the $OD_{600}$ of each sample was then measured in a plate reader (Synergy H1 microplate reader, Biotek). The samples were diluted to $OD_{600} = 0.036$ into fresh M9 media and sequentially diluted to $OD_{600} = 0.00036$ by adding 2 μl of the first dilution to 198 μl M9 media supplemented with inducers. Four combinations of inducer were tested for each tandem promoter (0 or 1 mM IPTG, and 0 or 20 ng/ml aTc). Cells were grown at 37°C at 1,000 rpm in an ELMI shaker for 8 h before samples were prepared for cytometry.

### Gate characterization

A single colony containing a plasmid carrying a gate was inoculated into 150 μl LB media in a V-bottom 96-well plate (Nunc, 249952) and grown at 37°C at 1,000 rpm in an ELMI shaker (ELMI, DTS-4) overnight (15–16 h). The overnight culture was diluted 200-fold into 200 μl M9 media in a V-bottom 96-well plate and grown at 37°C at 1,000 rpm in an ELMI shaker for 3 h. The $OD_{600}$ of each sample was then measured in a plate reader (Synergy H1 microplate reader, Biotek). The samples were diluted to $OD_{600} = 0.036$ into fresh M9 media and sequentially diluted to $OD_{600} = 0.00036$ by adding 2 μl of the first dilution to 198 μl M9 media supplemented with inducers. Cells were grown at 37°C at 1,000 rpm in an ELMI shaker for 8 h before samples were prepared for cytometry. To estimate the impact on growth rate, we measured the $OD_{600}$ at the end of the experiments (after 8 h of incubation) and normalized the values to the uninduced sample. Note that two input promoters were used to characterize the NOT gates to fully sweep through the upper range of inputs (not obtainable with a single input promoter). To do this, a tandem promoter ($P_{Tet}$ followed by $P_{Tac}$) was used. Each gate was characterized with eighteen inducer concentrations: 0 mM IPTG, 0 ng/ml aTc; 0.005, 0; 0.01, 0; 0.02, 0; 0.03, 0; 0.05, 0; 0.07, 0; 0.1, 0; 0.2, 0; 0.5, 0; 1, 0; 2, 0; 2, 0.5; 2, 1; 2, 2; 2, 5; 2, 10; and 2, 20. The response function for a gate has to have both the inputs and outputs in units of RPU. To convert the *x*-axis from inducer concentrations to RPU, a separate plasmid (JS_NOT_in, Appendix Fig S21) containing the tandem promoters driving *yfp* was constructed and grown under identical conditions. The fluorescence from this plasmid was measured for the same combinations of inducers, converted to RPU, and then used as the gate input for the response function.

### Measurement of gate dynamics

The following protocol was used to turn a gate off (output ON to OFF). A single colony was inoculated into 150 μl LB media in a V-bottom 96-well plate and grown at 37°C at 1,000 rpm in an ELMI shaker overnight (15–16 h). The overnight culture was diluted 200-fold into 200 μl fresh M9 media in a V-bottom 96-well plate and grown at 37°C at 1,000 rpm in an ELMI shaker for 3 h. The $OD_{600}$ of each sample was measured in a plate reader (Synergy H1 microplate reader, Biotek). The samples were diluted to $OD_{600} = 0.036$ into fresh M9 media and sequentially diluted to $OD_{600} = 0.00036$ by adding 2 μl of the dilution to 198 μl M9 media supplemented with inducers. Both the first and the second dilutions contained the necessary inducers to turn the two input promoters on: 0.1 mM IPTG for B1-BM3R1, C1-CymR, P1-PhlF, S1-SrpR, V1-VanR; 0.2 mM IPTG for F2-AmeR_2; 0.5 mM IPTG B3-BM3R1, P2-PhlF; 2 mM IPTG for S3-SrpR, S4-SrpR; 2 mM IPTG + 1 ng/ml aTc for A1-AmtR_2, P3-PhlF; and 2 mM IPTG + 2 ng/ml aTc for F1-AmeR_2, E1-BetI_2, B2-BM3R1, H1-HlyIIR_2, N1-LmrA_2, S2-SrpR. The cultures were grown at 37°C at 1,000 rpm in an ELMI shaker. Multiple wells were grown with each well corresponding to a time point (so that the entire sample can be removed and analyzed by flow cytometry). To obtain sufficient cells, early time points were grown in multiple (up to six) wells and then centrifuged (5 min at 4°C and 4,500 × *g* in Thermo Sorvall Legend XFR, 75004538) and combined prior to analysis.

The following protocol is for turning a gate on (output OFF to ON). A single colony was inoculated in 150 μl LB media supplemented with both inducers (concentration of both inducers for each gate is the same as above, used for gate output OFF) in a V-bottom 96-well plate and grown at 37°C at 1,000 rpm in an ELMI shaker overnight (15–16 h). The overnight culture was diluted 200-fold to 200 μl fresh M9 media with inducers in a V-bottom 96-well plate and grown at 37°C at 1,000 rpm in an ELMI shaker for 3 h. Cells were washed twice by centrifugation at 4°C at 4,500 × g for 5 min. The $OD_{600}$ of each sample was measured in a plate reader (Synergy H1 microplate reader, Biotek). The samples were diluted to $OD_{600}$ = 0.036 into fresh M9 media and sequentially diluted to $OD_{600}$ = 0.00036 by distributing 2 μl of the first diluted sample into 198 μl M9 media (neither containing inducers). The cultures were grown at 37°C at 1,000 rpm in an ELMI shaker. Multiple wells were grown with each well corresponding to a time point (so that the entire sample can be removed and analyzed by flow cytometry). To obtain sufficient cells, early time points were grown in multiple (up to six) wells and then centrifuged (5 min at 4°C and 4,500 × g in Thermo Sorvall Legend XFR, 75004538) and combined prior to analysis.

### Measurement of sensor dynamics

The sensors were measured following the protocol for gates (above). The concentrations of inducers to turn the sensors on were as follows: 1 mM IPTG (LacI); 20 ng/ml aTc (TetR); 2 μM OC6 (LuxR); and 2 μM OHC14 (CinR).

### User constraint file

Cello uses a file, referred to as the UCF, that contains empirical gate data needed to design circuits. A new UCF (Eco1C2G2T2) was created from this work and provided as Appendix Data. The UCF contains the gate technology and data (response functions and $OD_{600}$ measurements). It also defines the strain, genetic location of the circuit, and the growth conditions where the circuit design predictions are valid. Eco1C2G2T2 is based on E. coli strain NEB 10-beta, and the location of the circuits is the p15a plasmid JS_BB_2 (Appendix Fig S17). The specification for the growth conditions differs slightly from UCF Eco1C1G1T1 (Nielsen et al, 2016). Instead of performing a 650-fold dilution, we set the starting point for the circuit measurement based on cell density ($OD_{600}$ = 0.00036) and the growth time is longer (8 h). Gates were added (F1-AmeR_2, F2-AmeR_2, A1-AmtR_2, E1-BetI_2, H1-HlyIIR_2, and N1-LmrA_2; see section above), and I1-IcaR(A), R1-PsrA, Q1-QacR, and Q2-QacR were removed because we often observed them to produce circuits that slowed growth (not shown). Additionally, L1-LitR was removed because this gate was found to not be sufficiently orthogonal (not shown). All of the parameters describing the gates are included in a single UCF file, including those capturing roadblocking and dynamics. Eugene rules (Oberortner et al, 2014) are also included to specify the organization of gates onto a linear DNA sequence. This UCF uses the following layout rules for gates and Type IIS cloning scars. A specific gate order is enforced (forward orientation): VanR—PhlF—SrpR—AmtR—AmeR—BM3R1—LmrA—HlyIIR—BetI—CymR. A pair of gates are not sufficiently orthogonal (CymR:SrpR), so rules were included prohibiting their use simultaneously in a circuit. Similarly, the use of multiple gates that use the same repressor and only differ

via their RBSs is prohibited from appearing in the same circuit. Cloning scars appear in the order A—B—D—E—F—X—V—U—C with A/C always appearing at the left and right most positions and B-U appearing as needed (added starting from B). Rules from UCF Eco1C1G1T1 prohibiting roadblocking promoters in the downstream input position were removed.

### Computational circuit design

The Cello software was used to design the circuit DNA sequences. The code was changed in several ways, as compared to the previously published version (Nielsen et al, 2016). The new code is available at http://github.com/CIDARLAB/Cello-v2 and runs both the old and new UCF. The instructions to install and run the Cello code are available at http://www.cellocad.org/about.html. The code and UCF format were modified to allow for the calculation of non-additive tandem promoters as the input to a gate. This calculation is part of gate assignment and calculation of growth impact. When the UCF is loaded, the new code looks for the parameters for the non-additive version of the promoter model (written in the UCF as "[{"name": "alpha", "value": 0.24070015},{"name": "beta", "value": 0.061908386}]"). If these are absent, then the code reverts to using the additive model as described previously. All of the circuits in this manuscript are based on the Eco1C2G2T2 UCF and the default parameters unless otherwise specified. Cello can accept Verilog files that specify the logic function (truth table) or structural Verilog where the complete wiring diagram is defined. The A, C, F, and G circuits were specified as truth tables, and Cello identified the wiring diagram using logic minimization. In the case of segments B, D, and E, the circuits were identified using enumeration and then the wiring diagram specified in Cello using structural Verilog. Sensor data were also provided to Cello, including the DNA sequences of the output promoters and the RPU values associated with the OFF and ON states of each sensor. The values used are (OFF/ON): $P_{Tac}$ 0.008/1.686 RPU (± 0.2 mM IPTG), $P_{Tet}$ 0.04/1.967 RPU (± 2 ng/ml aTc), $P_{Lux2}$ 0.03/2.234 RPU (± 0.1 μM OC6), and $P_{Cin}$ 0.005/3.178 RPU (± 1 μM OHC14). The output of Cello includes DNA sequences for the circuit, which were constructed as specified, and predictions for cytometry fluorescence data and the impact on growth.

### Circuit characterization

The circuits were constructed based on the DNA sequence specified by Cello and inserted into the same p15a plasmid backbone specified in the UCF (JS_BB_2). The circuit output promoter(s) were carried on a separate higher-copy pSC101 plasmid (var2, JS_BB_3; Appendix Fig S17) and transcriptionally fused to yfp expressed with the same RBS/5′-UTR as used for gate characterization (Segall-Shapiro et al, 2018). This plasmid was selected because it has a nearly identical copy number with the p15a plasmid carrying the circuit (ratio of 1.05) (Segall-Shapiro et al, 2018), so there is no need to scale the experimentally measured fluorescence values to those predicted by Cello. The maps for the circuit/output plasmids are shown in Appendix Fig S24, and the annotated circuit DNA sequences are provided in Appendix Table S5. Circuit characterization experiments were performed as described for "Gate characterization" (above). The inducers were prepared as combinations of 0.2 mM IPTG, 2 ng/ml aTc, 0.1 μM OC6, and 1 μM OHC14.

## LCD display

To visualize the "LCD Display", we designed a 3D-printed scaffold that contains copies of the seven segment patterns. The dimensions of this device are 56 × 36.4 × 6 mm (Fig 3A and Appendix Fig S16). Chambers that are round-trimmed serve to depict each segment. The dimensions of each bar corresponding to a segment are 5.6 × 1.4 × 3 mm. The pattern was designed with CAD software (Autodesk Fusion 360), and the object was printed using Nylon PA12 for the device material (shapeways.com). For demonstrating the display, the same procedure is followed for circuit characterization above, except replacing the V-bottom 96-well plate (Nunc, 249952) with a 2 ml 96-well deep well plate (USA scientific, 1896–2000) and using a multitron (INFORS HT; 37°C and 900 rpm) instead of an ELMI shaker. After 8 h of incubation, the $OD_{600}$ was measured using a plate reader and converted using the equation: $CFU/ml = 2 \times 10^9$ (measured $OD_{600}$)$^{0.9759}$. The samples were diluted so that the number of cells is $0.86 \times 10^9$ in 20 μl of the PBS/Kan mixture described for the cytometry experiments (above). We scaled the number of cells added to each segment of the LCD display based on the RPU associated with the ON state, noting that the magnitude of the ON states for each circuit is quantitatively different (see caption for details). We added 20 μl of those samples into each chamber of a 3D-printed scaffold. The photo was taken using a Chemidoc XRS+ system (Bio-Rad) with CCD camera and 530/28 nm filter. For publication, the entire picture was processed equality by adjusting the brightness and contrast using Adobe Photoshop CC 2019.

## Circuit time courses

Two time courses were performed and combined to produce the trajectories shown in Fig 3B. The first was a continuous experiment where the circuit was passaged for 88 h through all of the states, progressing serially from 0 to 9. To change the digit, an aliquot is taken to measure the fluorescence, the cells were spun down and resuspended in fresh media with the new inducers, and then, the cells are grown again. This leads to one data point every 8 h (large colored points in Fig 3B). To obtain additional data connecting these points, separate experiments are performed focusing on only one digit change and data points are taken every 2 h (small black points in Fig 3B). This was done to avoid continuous human intervention over 88 h at 2-h resolution. Details are provided below.

*Continuous switching experiments* ($0 \rightarrow 1 \rightarrow 2 \rightarrow 3 \rightarrow 4 \rightarrow 5 \rightarrow 6 \rightarrow 7 \rightarrow 8 \rightarrow 9 \rightarrow 0$). The circuits were initiated by growing cultures as described in "Circuit characterization" (above) without inducers (the 0000 state). Cells were washed twice by centrifugation (Thermo Sorvall Legend XFR, 75004538, 4°C, 4,500 × g for 5 min) and resuspended with 200 μl M9 media supplemented with 0.1 μM OC6 (the 0001 state). The $OD_{600}$ was measured, and the samples were first diluted to $OD_{600}$ = 0.036 into fresh M9 media (0.1 μM OC6) and then 1:100 into 200 μl M9 media (0.1 μM OC6) (final $OD_{600}$ = 0.00036). The same procedure is sequentially repeated by changing inducers.

*Independent digit-to-digit experiments* (separate experiments: 0→1, 1→2, 2→3, 3→4, 4→5, 5→6, 6→7, 7→8, 8→9, and 9→0) A single colony was picked into 20 μl LB media, of which 1 μl was distributed into 10 wells, each of which contained 150 μl LB media

and inducers (V-bottom 96-well plate). Each well contained the combination of inducers (see "Circuit characterization", above) corresponding to one of the ten digits. The cultures were incubated at 37°C at 1,000 rpm in an ELMI shaker overnight (15–16 h). The overnight cultures were then diluted 200-fold into 200 μl M9 media supplemented with the same combination of inducers corresponding to their starting digit states in a V-bottom 96-well plate. These cultures were incubated at 37°C at 1,000 rpm in an ELMI shaker for 3 h. The cells were then washed twice by centrifugation (Thermo Sorvall Legend XFR, 75004538, 4°C, 4,500 × g, 5 min). Finally, the pellets were resuspended with 200 μl M9 media containing the set of inducers for the next consecutive digit (e.g., the culture that was started as 0 is now induced to become 1). The $OD_{600}$ was then measured and diluted to 0.00036 into M9 containing inducers, as described for the continuous experiments, above. Multiple wells were grown with each well corresponding to a time point (so that the entire sample can be removed and analyzed by flow cytometry). To obtain sufficient cells, early time points were grown in multiples (up to four) wells and then centrifuged (5 min at 4°C and 4,500 × g in Thermo Sorvall Legend XFR, 75004538) and combined prior to analysis.

## Analysis of broken circuits

After the 88-h time course (above), the circuit characterization experiments were repeated as described above. Even though the correct digit was shown at every 8-h time point up to 88 h, this is only a subset of all of the states of each circuit. The cells containing segment F failed for some states after the 88-h time course. The plasmid sequence was evaluated to identify mutations. To do this, we spread a small amount of the last culture onto a fresh LB-agar plate with 50 μg/ml kanamycin. After incubating for 24 h at 37°C, we picked four colonies for sequencing analysis. All four DNA sequences for the circuit were correct; however, three clones were disrupted through the insertion of either an IS5-like or IS4-like transposable element either in the *luxR* gene or in the promoter, BBa_J23104 used for LuxR and CinR expression. All four plasmids were transformed into a fresh *E. coli* strain NEB 10-beta containing the output plasmid, and the circuit characterization experiments were repeated. One colony which was not disrupted by the transposable element insertion functioned correctly (Appendix Fig S13).

## Analysis of initial circuit designs (on high copy plasmids), including broken states

The 7-segment circuits were initially designed on a high copy plasmid (Appendix Fig S18). The 88-h time course was repeated four times. All four clones of segments D, F, and G; three clones of segment B; and one clone of segment C showed failed states before finishing the experiment (Appendix Fig S14). Of these, segments F and G were further analyzed to determine the cause of failure. After the 88-h time course, PCR was used to amplify the circuit DNA, including the reporter (from scar A to scar C). Surprisingly, when this region was sequenced, it had the correct sequence. This could indicate that there is a mixed population of plasmids. To determine this, we spread an aliquot of the culture after the 88-h experiment

onto a fresh LB-agar plate containing 50 µg/ml kanamycin. After incubating for 24 h at 37°C, we picked four colonies, grew them in LB liquid culture with 50 µg/ml kanamycin, and extracted plasmids via miniprep. We then transformed these plasmids into a fresh *E. coli* strain NEB 10-beta, and the circuit characterization experiments were repeated. Three colonies of segment F and one colony of segment G recovered functioned normally whereas the others failed (Appendix Fig S15).

**Circuit dynamics simulation using MATLAB**

Ordinary differential equations were derived to model gate dynamics (Appendix Informations II and III). MATLAB scripts were written to extract parameters from the empirical data using the ODE solver ODE15s. The script "input_ontoff" was used to estimate the YFP degradation rate from time course experiment of YFP expression driven by an input promoter used for gate characterization (ON to OFF). The script "input_offtoon" was used to estimate the induction parameter for sensors based on the time courses experiment of YFP expression by sensor promoters (OFF to ON). The script "NOTgates_fitting" extracts induction and relaxation parameters for the gates based on the time courses (ON to OFF and OFF to ON). To simulate the circuit dynamics shown in Fig 3B, the script "Solver" was used in conjunction with the function script "Equations". Solver solves the time trajectory for each output promoter using ODE15s. Equations used multiple times in Solver when new inputs are required to switch states (every 8 h), and inputs were entered as either OFF or ON value: 0.008 or 1.686 RPU, 0.04 or 1.967 RPU, 0.03 or 2.234, and 0.005 or 3.178 RPU for $P_{Tac}$, $P_{Tet}$, $P_{Lux2}$, and $P_{Cin}$, respectively. Solver contains the Heaviside step function to switch the induction and relaxation parameters depending on the steady state at each time point. The Heaviside step function ($H(z)$) is defined as 0 when $z$ is negative and as 1 when $z$ is positive. All MATLAB scripts are available on GitHub (https://github.com/VoigtLab).

# Data availability

The dataset and computer code produced in this study are available in the following databases:

- UCF information (Eco1C2G2T2.UCF, Eco1C2G2T2.input, and Eco1C2G2T2.output) is available in Dataset EV1.
- Codes used to process the data: GitHub (http://github.com/CIDARLAB/Cello-v2 and https://github.com/VoigtLab).

Expanded View for this article is available online.

## Acknowledgements
This work was supported by Office of Naval Research, Multidisciplinary University Research Initiative grant N00014-16-1-2388 (C.A.V., J.S., B.D., A.A.K.N., and S.Z.); US National Institute of General Medical Sciences (NIGMS) P50 GM098792; National Science Foundation (NSF) CCF-1807575; and DARPA SD2 FA8750-17-C-0229 (C.A.V.).

## Author contributions
JS designed and performed the experiments and the computational models, analyzed data, and wrote the manuscript. SZ performed computational work.
AAKN and BSD performed the initial circuit design. CAV designed experiments, analyzed data, and wrote the manuscript.

## Conflict of interest
The authors declare that they have no conflict of interest.

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
