## [Review Process File · Molecular Systems Biology]

Programming Escherichia coli to function as a digital display

Christopher Voigt, Jonghyeon Shin, Shuyi Zhang, Bryan Der and Alec Nielsen

Review timeline:	Submission date:	9 th December 2019
	Editorial Decision:	7 th January 2020
	Revision received:	4 th February 2020
	Accepted:	7 th February 2020

Editor: Maria Polychronidou

Transaction Report:

1st Editorial Decision

7th January 2020

Thank you again for submitting your work to Molecular Systems Biology. We have now heard back from the two referees who agreed to evaluate your study. Overall, the reviewers are quite supportive. They raise however a series of (mostly minor) concerns, which we would ask you to address in a revision. The recommendations of the reviewers are rather clear and I think that there is no need to repeat any of the points listed below.

REFeree REPORTS

Reviewer #1:

Summary and general comments

The authors developed an updated model for the synthetic gene circuit modeling software Cello. They update their model to account for the phenomenon of "RNAP roadblocking," which ultimately enables better prediction of gate function. The authors then utilize their updated model to design a large genetic circuit function as a binary-coded digit to 7-segment decoder, which functions as predicted. The scope of in-depth parts characterization and circuit design is impressive and is a model to show how modeling and experiments in synthetic biology can be used to achieve complex design goals. The manuscript also highlights the importance of updating models to better reflect the underlying biology in order to be able to design complex biological processes. The manuscript is generally well written, and data and methods are clearly explained. The updated model is well explained with detailed derivations and appears to be mathematically correct. I commend the authors on their thorough work and recommend the manuscript be published after addressing the several comments below.

Major comments:

- How does the size and complexity of this circuit compare to others that have been previously published? The authors discuss that the complexity is in the range that requires design automation, but do not put the circuit size in the context of the broader literature. It appears to be one of the largest implemented to date, though discussing the size of the circuit in each of the 7 strains may be the more relevant size comparison. While the scale of the design is impressive, and the fact that all 7 gates work well is a feat in and of itself, it appears to be 7 essentially independent design problems and is not in fact a "63-regulator network" as stated in the abstract since many of these regulators are not actually connected to one another. The language in the manuscript should be changed to more accurately reflect the design realities.
- Additional discussion of what remains to be improved to further increase accuracy of the model would add greater context to the improvements made (e.g. improved R2 on prediction from 0.45 to 0.76, but how can it be improved further?). When will the improved model break down?
- While the authors provide DNA and plasmid diagrams for all of the reported circuits, annotated GenBank files should be provided as part of the supplement. This would also allow others to replicate their work.
- Greater discussion of the utility and response times of large genetic circuits is warranted.
- The authors indicate that different numbers of cells were induced in each segment in order to achieve a similar signal from each segment due to the difference in output of each designed circuit. While this allows them to take a nice picture (Figure 3A), this may mislead readers into thinking that the output is more uniform across the different circuits than it really is. This needs to be better discussed.

Minor comments:

- It appears that Table 1 is missing some of the relevant units, particularly for the characteristic response times. These should be added.
- Figure 2B, the image of a Texas Instruments chip does not add much conceptually to the paper. I would recommend removing it or better folding in why the image is there.
- Some of the figures in the supplement contain elements that are quite small. Given that there is a significant amount of data associated with the manuscript, this is expected, but it can be difficult to read when zoomed in because it does not appear that the figures are vector graphics as some of the small text becomes pixelated and difficult to read upon zooming. Please increase the size so that it is legible.
- On page 7 paragraph 2, the last sentence appears to be incomplete. "The results of running Cello with the previous UCF..." It seems to point to Supplementary Figure S9, but does not offer interpretation. Can an interpretation be added and the sentence completed.
- On page 8, the sentence starting "Ultimately..." is not a grammatically correct sentence. Same for page 9, the sentence starting "For example..."
- On page 9, the authors state that it would be impactful for genetic engineering projects if a tool existed which could predict the genetic stability of synthetic systems. Such a tool does exist which meets some of these needs: see and cite "Predicting the genetic stability of engineered DNA sequences with the EFM calculator"
- There are several locations in the manuscript where it is stated that predictions match experiments "closely" (page 7 discussion of population data matching cytometry and page 8 discussion of time course). While it is obvious by visual inspection that the claim is reasonably true, a more quantitative analysis would add more weight to the claim. Please add.

Reviewer #2:

Summary

- Describe your understanding of the story

This is a short paper reporting the construction of a 7-segment digital display that harkens back to the early days of consumer digital electronics, using genetic circuits in E coli. It describes an ambitious exploration of the state of the art of the gate technology and design methodologies being developed in the Voigt lab. It advances their promising efforts to open a universe of new applications, and it also serves as a cautionary note that there remains work to do.

- What are the key conclusions: specific findings and concepts

This paper is proof that, with sufficient care, elaborate digital logic circuits (e.g. seven different circuits of up to eight gates each) can be designed automatically by software and implemented in cells using modular parts (sensors and gates). In a sense, a core dream of modern synthetic biology has thus been achieved, pushing it beyond manual parts "hacking." However, the road was apparently bumpier than was anticipated in 2016 ("Genetic circuit design automation"), and the paper is laudably honest in this regard. Success apparently required theoretical development (new gate model including roadblocking interactions) and experimental protocols to better characterize gates, amongst other innovations. It appears likely that a process of innovation and refinement will continue to be required in order to fulfill the promise of the technology.

- What were the methodology and model system used in this study

This study used an improved version of the formalism and library of genetic parts announced earlier by the Voigt lab (Nielsen et al, Genetic circuit design automation, Science 2016). The seven segments of the digital display employed seven different strains of E coli, which contained low- or high-copy plasmids carrying the genetic circuits that implemented the relevant logic. The input to the device was encoded in binary by the presence or absence of inducer molecules.

General remarks

- Are you convinced of the key conclusions?

The key conclusions are well-described. The justification of the model and the characterization results are sufficient. Replication of 88 hour time courses exercising the logic over different days was performed. Failure modes were identified and dealt with (for the current purposes). A few failures (segment F) were still observed in some replicates.

- Place the work in its context.

Due to fluctuations and noise, it will always be somewhat a matter of judgment (e.g. application purpose) to say whether a circuit/device "fails" or not. This study mentions the use of thresholds and objective functions. More could be said on the applicability and maturity of the technology for different application scenarios, e.g. detection versus biomedical use. Now that this device exists, the key takeaways could be more clearly described to a wider audience.

- What is the nature of the advance (conceptual, technical, clinical)?

A key point is that modular NOR gates, as implemented in the current repressor logic, are not simple to make and involve trade-offs. The development of a roadblocking-aware model with two extra parameters alpha and beta was apparently critical, as were experimental protocols to optimize and characterize parts given the new model. The device time courses set a new baseline for others working in this space who may wish to optimize/improve on different aspects.

- How significant is the advance compared to previous knowledge?

This study is where the rubber meets the road. It is a great accomplishment, while also demonstrating that what looks easy may not be in practice. It was important that this study be done.

- What audience will be interested in this study?

All readers interested in synthetic biology will find something of interest. It is likely to have high popular appeal and draw in readers from different backgrounds, e.g. electrical engineering.

Major points

-Specific criticisms related to key conclusions

No major criticisms. The time course dataset could, or perhaps should, be made available.

-Specify experiments or analyses required to demonstrate the conclusions

The paper can do a better job of describing the takeaways from this study.

- Would different Cello algorithms have modified the results, and is algorithm development still a strategic issue?

- Is maintaining orthogonality/non-homology an essential bottleneck?

- If a user requires state switching on faster time scales, what are the options?

- What are the most salient trade-offs or constraints between noise, speed, RNAP flux/load, evolution?
- What is the "device lifetime"?
- Is the technology ready for prime time, and can it be used in a biomedical context?
- Is the basal and median constitutive load a key characteristic that should be reported for parts and circuits; can the scheme scale?
- Many of the alpha and beta parameter values in Table 1 differ considerably from the ideal of 1 (and they are possibly anti-correlated). Comments?
- Would be nice to have some comments on the device operating conditions. How does it respond to perturbations, pH, temperature, light, growth, etc.

-Motivate your critique with relevant citations and argumentation

Minor points

-Easily addressable points

- As the title of the paper is meant to draw in diverse readers, a short primer on the gate technology and logic conventions should be included in a Supplement. This is needed to orient readers who are not aware of the lab's recent work. Possible topics: the class distinction between sensors and gates; the fact that presence of a molecule signifies 1 in the former and the absence signifies 1 in the latter; RNAP flux is akin to voltage; the output promoter of one gate is the input promoter for the next, which violates one notion of modularity/independence; a gate need not be a single physical entity but could have the output promoter in trans. A worked example of two NOT gates in series could help fix concepts.

- The first sentence of the abstract states, "Synthetic genetic circuits offer the potential to wield computational control over biology, but their complexity is limited by the accuracy of mathematical models." This gives the impression that loading Cello with better models is the only barrier to further complexity. This sentence should be modified.

- The non-monotonic behavior observed at high input levels in Figure S8 deserves comment.

- What are the bacteria doing? Are they stressed?

-Presentation and style

-Trivial mistakes

These sentences are ungrammatical, requiring an extra word and/or rephrasing.

- "The results of running Cello with the previous UCF (Eco1C1G1T1) (Supplementary Figure S9)."
- "Ultimately, we decided to restart the project and redesigned the underlying gates used them to rebuild the circuits reported in this manuscript."
- "For example, they could be encoded at low copy in genome, high-affinity repressors reduce the amount that have to be expressed, and using smaller repressors built amino acids that have lower material and energy requirements." Maybe use bullet points?

1st Revision - authors' response

4th February 2020

Reviewer #1:

1. *How does the size and complexity of this circuit compare to others that have been previously published? The authors discuss that the complexity is in the range that requires design automation, but do not put the circuit size in the context of the broader literature. It appears to be one of the largest implemented to date, though discussing the size of the circuit in each of the 7 strains may be the more relevant size comparison. While the scale of the design is impressive, and the fact that all 7 gates work well is a feat in and of itself, it appears to be 7 essentially independent design problems and is not in fact a "63-regulator network" as stated in the abstract since many of these regulators are not actually connected to one another. The language in the manuscript should be changed to more accurately reflect the design realities.*

We have edited the language, as suggested.

2. *Additional discussion of what remains to be improved to further increase accuracy of the model would add greater context to the improvements made (e.g. improved R2 on prediction from 0.45 to 0.76, but how can it be improved further?). When will the improved model break down?*

We believe that the model is close to what is possible regarding the modeling of the roadblocking phenomena. More accurate modeling will require the re-design of the gate itself, moving from plasmid-based systems to the genome, and reducing the resource requirements of the gates. We have included this information in the discussion.

3. *While the authors provide DNA and plasmid diagrams for all of the reported circuits, annotated GenBank files should be provided as part of the supplement. This would also allow others to replicate their work.*

We have placed the plasmids in Genbank.

4. *Greater discussion of the utility and response times of large genetic circuits is warranted.*

The response times are quite slow, 2-4 hours. However, when programming a dynamic circuit where intermediate wires are used to control different processes, a series of responses can be timed with 30-60 minute resolution. In terms of fermentation, which can take place over days, this is sufficient to alter enzyme expression in order to coordinate the timing with each other or with shifts in environmental conditions. Clearly, faster circuits would be beneficial, but ultimately if the output requires genes be turned on or off and translated/degraded, these timescales will be on the order of hours.

5. *The authors indicate that different numbers of cells were induced in each segment in order to achieve a similar signal from each segment due to the difference in output of each designed circuit. While this allows them to take a nice picture (Figure 3A), this may mislead readers into thinking that the output is more uniform across the different circuits than it really is. This needs to be better discussed.*

We have carefully added and edited text so that this point is clear.

6. *It appears that Table 1 is missing some of the relevant units, particularly for the characteristic response times. These should be added.*

They have been added.

7. *Figure 2B, the image of a Texas Instruments chip does not add much conceptually to the paper. I would recommend removing it or better folding in why the image is there.*

It has been removed.

8. *Some of the figures in the supplement contain elements that are quite small. Given that there is a significant amount of data associated with the manuscript, this is expected, but it can be difficult to read when zoomed in because it does not appear that the figures are vector graphics as some of the small text becomes pixelated and difficult to read upon zooming. Please increase the size so that it is legible.*

The size has been increased.

9. *On page 7 paragraph 2, the last sentence appears to be incomplete. "The results of running Cello with the previous UCF..." It seems to point to Supplementary Figure S9, but does not offer interpretation. Can an interpretation be added and the sentence completed.*

This has been edited.

10. *On page 8, the sentence starting "Ultimately..." is not a grammatically correct sentence. Same for page 9, the sentence starting "For example..."*

The change has been made.

11. *On page 9, the authors state that it would be impactful for genetic engineering projects if a tool existed which could predict the genetic stability of synthetic systems. Such a tool does exist which meets some of these needs: see and cite "Predicting the genetic stability of engineered DNA sequences with the EFM calculator"*

The reference has been added.

12. *There are several locations in the manuscript where it is stated that predictions match experiments "closely" (page 7 discussion of population data matching cytometry and page 8 discussion of time course). While it is obvious by visual inspection that the claim is reasonably true, a more quantitative analysis would add more weight to the claim. Please add.*

We have added a figure to the Appendix that shows the fit of the prediction to experimental data, including R² values.

Reviewer #2:

1. *Would different Cello algorithms have modified the results, and is algorithm development still a strategic issue?*

There are two levels to the Cello algorithm: 1. the logic minimization step, and 2. the assignment of specific repressors to each NOR gate. The first algorithm is based on logic minimization algorithms that have emerged from Electrical Engineering over decades of research. As with electronics, we would benefit from better algorithms, particularly if they could be tuned to solving the

problem with respect to the needs of genetic circuit design (e.g., limited to limited permutations of specified gate types). The second step is based on simulated annealing and is not a bottleneck as the repressor assignment problem is not particularly challenging computationally. It would benefit from gates that are easier to put together (i.e., their response functions “fit” together more easily).

2. *Is maintaining orthogonality/non-homology an essential bottleneck?*

Yes, it is critical.

3. *If a user requires state switching on faster time scales, what are the options?*

There are not any good ones. Ultimately, if the output is the expression of a gene (as opposed to, say, chemotaxis), then turning on and off the gene is going to be limited by transcription/translation and protein/mRNA degradation. The transcriptional circuits we present are on this same timescale. For faster responses that do not involve gene expression, one can imagine custom circuits based on phosphorylation or protein-protein interactions.

4. *What are the most salient trade-offs or constraints between noise, speed, RNAP flux/load, evolution?*

The only trade-off in the above is between RNAP load and evolution, which we discuss in the paper. Noise and speed are not related to this or each other.

5. *What is the “device lifetime”?*

It depends on the RNAP/ribosome usage, which changes depending on the device state, which we discuss in the paper.

6. *Is the technology ready for prime time, and can it be used in a biomedical context?*

No and no. The biggest challenges are to move away from plasmid-based systems and build circuits in the genome, build UCFs for biomedically-related organisms (perhaps the closest would be *E. coli* Nissle), and reduce the growth impact of the circuit so as to not affect titer or viability in a complex environment. These are all areas being actively researched.

7. *Is the basal and median constitutive load a key characteristic that should be reported for parts and circuits; can the scheme scale?*

Yes, although it is dependent on state, which makes it more complex than a single number. It is also not easy to quantify, making it difficult to capture and report for large part libraries.

8. *Many of the alpha and beta parameter values in Table 1 differ considerably from the ideal of 1 (and they are possibly anti-correlated). Comments?*

A value of 1 is not “ideal,” rather it means “no roadblocking.” The divergence from 1 indicates that roadblocking is an issue for these promoters.

9. *Would be nice to have some comments on the device operating conditions. How does it respond to perturbations, pH, temperature, light, growth, etc.*

The device operating conditions depend on the specific conditions defined in the UCF. To the extent that there are tolerances to, say pH, we do not know. It requires testing the circuit under those conditions. We do not have a framework for capturing or predicting it generically.

10. *As the title of the paper is meant to draw in diverse readers, a short primer on the gate technology and logic conventions should be included in a Supplement. This is needed to orient readers who are not aware of the lab's recent work. Possible topics: the class distinction between sensors and gates; the fact that presence of a molecule signifies 1 in the former and the absence signifies 1 in the latter; RNAP flux is akin to voltage; the output promoter of one gate is the input promoter for the next, which violates one notion of modularity/independence; a gate need not be a single physical entity but could have the output promoter in trans. A worked example of two NOT gates in series could help fix concepts.*

The text has been edited to include better definition of the terms. We have declined to include a “primer” in the SI, instead referring to earlier published work.

11. *The first sentence of the abstract states, "Synthetic genetic circuits offer the potential to wield computational control over biology, but their complexity is limited by the accuracy of mathematical models." This gives the impression that loading Cello with better models is the only barrier to further complexity. This sentence should be modified.*

The change has been made.

12. *The non-monotonic behavior observed at high input levels in Figure S8 deserves comment. What are the bacteria doing? Are they stressed?*

This is an observation that we make frequently. Yes, this is a region where cells are slowing in growth, thus decreasing the protein degradation rate and causing the apparent increase in fluorescence. We ignore these data in the fit to the response function.

13. *These sentences are ungrammatical, requiring an extra word and/or rephrasing. "The results of running Cello with the previous UCF (Eco1C1G1T1) (Supplementary Figure S9)." "Ultimately, we decided to restart the project and redesigned the underlying gates used them to rebuild the circuits reported in this manuscript." "For example, they could be encoded at low copy in genome, high-affinity repressors reduce the amount that have to be expressed, and using smaller repressors built amino acids that have lower material and energy requirements." Maybe use bullet points?*

We have edited these sentences for clarity.

Accepted

7th February 2020

Thank you again for sending us your revised manuscript. We are now satisfied with the modifications made and I am pleased to inform you that your paper has been accepted for publication.

Corresponding Author Name: Christopher A. Voigt

Manuscript Number: MSB-19-9401